# A Dual-Cavity Fiber Fabry–Pérot Interferometer for Simultaneous Measurement of Thermo-Optic and Thermal Expansion Coefficients of a Polymer

**DOI:** 10.3390/polym14224966

**Published:** 2022-11-16

**Authors:** Cheng-Ling Lee, Chao-Tsung Ma, Kuei-Chun Yeh, Yu-Ming Chen

**Affiliations:** 1Department of Electro-Optical Engineering, National United University, Miaoli 360, Taiwan; 2Department of Electrical Engineering, National United University, Miaoli 360, Taiwan

**Keywords:** dual-cavity fiber Fabry–Pérot interferometer (DCFFPI), thermo-optic coefficient (TOC), thermal expansion coefficient (TEC), polymer

## Abstract

This paper presents a novel method based on a dual-cavity fiber Fabry–Pérot interferometer (DCFFPI) for simultaneously measuring the thermo-optic coefficient (TOC) and thermal expansion coefficient (TEC) of a polymer. The polymer is, by nature, highly responsive to temperature (T) in that its size (length, L) and refractive index (RI, n) are highly dependent on the thermal effect. When the optical length of the polymer cavity changes with T, it is difficult to distinguish whether there is a change in L or n, or both. The variation rates of L and n with a change in T were the TOC and TEC, respectively. Therefore, there was a cross-sensitivity between TOC and TEC in the polymer-based interferometer. The proposed DCFFPI, which cascades a polymer and an air cavity, can solve the above problem. The expansion of the polymer cavity is equal to the compression of the air cavity with the increase in T. By analyzing the individual optical spectra of the polymer and air cavities, the parameters of TOC and TEC can be determined at the same time. The simultaneous measurement of TOC and TEC with small measured deviations of 6 × 10^−6^ (°C^−1^) and 3.67 × 10^−5^ (°C^−1^) for the polymer NOA61 and 7 × 10^−6^ (°C^−1^) and 1.46 × 10^−4^ (°C^−1^) for the NOA65 can be achieved. Experimental results regarding the measured accuracy for the class of adhesive-based polymer are presented to demonstrate the feasibility and verify the usefulness of the proposed DCFFPI.

## 1. Introduction

An acrylic-based polymer can be developed into many kinds of optical adhesives that can be widely utilized in various technologies and industrial applications. There are various highly transparent polymer materials that can be employed as functional optical materials for applications in the optics field. However, there are few studies on the investigation and simultaneous measurement of the optical or mechanical parameters of the polymers. The optical parameter, the thermo-optic coefficient (TOC), and the mechanical parameter, the thermal expansion coefficient (TEC), are both very important values because they reveal how the thermal effect would affect the properties of optical materials. As the optical materials in optoelectronic devices are greatly influenced by the thermal effect of the surrounding environment, the parameters of TOC and TEC have to be precisely determined simultaneously in practical applications. However, studies on sensing these two parameters simultaneously are very rare. In 2017, a low-cost monochromatic single-arm double interferometer was developed for the simultaneous measurement of the linear thermal expansion and thermo-optic coefficients of transparent samples [1]. Similar work published in the literature on the simultaneous analysis of the optical and mechanical properties of polymers has shown the importance of the issues investigated in this paper [2,3,4]. Hossain et al. reported a simple method for the simultaneous measurement of the thermo-optic and stress-optic coefficients of polymer thin films using a prism coupler technique with two different kinds of polymers. The finite element method was used to predict the stresses of the polymer film to further obtain its real thermo-optic coefficient [2]. Furthermore, Shimamura, A., et al. presented the simultaneous analysis of the optical and mechanical properties of cross-linked azobenzene-containing liquid–crystalline polymer films. The photoinduced stress of the polymer films upon UV irradiation was investigated to gain a fundamental understanding of the complicated physical processes of the photomechanical response [4]. The above-mentioned methodology and implementation mechanisms are all effective; however, the measurement systems used were general bulk, which could limit their further application. In recent decades, fiber optical sensors have been developed rapidly because of their excellent sensing performances and capability to function under variable conditions. In the measurement of the TOC or TEC of materials using fiber-based sensors, Esposito et al. proposed a fiber Bragg grating (FBG) sensor to measure the TEC of polymers at cryogenic temperatures [5]. Other work has proposed a fiber-optic temperature sensor based on the difference in TEC between fused silica and metallic materials [6]. The above studies are the results of TEC measurement by fiber sensors. Regarding the precise measurement of TOC, the information regarding the temperature (T) and refractive index (RI) variations enables the TOC of materials to be accurately measured by the following fiber-optic sensors: in-line hollow-core fiber Fabry-Pérot interferometers (HCFFPIs) [7,8], FBG [9], the core-offset fiber Mach–Zehnder interferometer (FMZI) [10] and two-mode fiber interferometer (TMFI) [11]. However, the above structures are still unable to measure the parameters of TEC and TOC concurrently. One solution is to take multiplexing advantage of the fiber-optic sensors. In view of the distributed and multiplexed capabilities of the fiber-based sensors, simultaneous measurement would be the most attractive advantage, as multiple fiber sensing elements can be spliced and integrated in a single-fiber device [12,13,14,15,16,17,18,19], e.g., a fiber Fizeau interferometer (FZI) connected to a FBG for the measurement of the relative humidity (RH) and T [12], two long-period fiber gratings (LPGs) for strain (S) and T measurement [13], a LPG/photonic crystal fiber interferometer (PCFI) for sensing the T and RI [14], two polymer FZIs for measuring the RH and T [15], a hybrid structured fiber Fabry–Perot interferometer (FFPI) for sensing the S and T simultaneously [16], a fiber Michelson interferometer for wide-range curvature measurement with low T cross-sensitivity [17], a single optical fiber tip modified with a coating of poly(allylamine hydrochloride) (PAH)/silica nanoparticles (SiO_2_ NPs) for RH measurement and then coated with thermochromic liquid crystal (TLC) for sensing T to achieve the simultaneous measurement [18], and a pendant polymer droplet-based fiber Fabry–Pérot interferometer (FFPI) for sensing the pressure and T [19]. The above sensing results based on multiple fiber sensing elements demonstrated the superiority of sensing characteristics in eliminating the possible cross-sensitivities problems; however, studies focused on the simultaneous sensing of the TOC and TEC are actually very limited.

In this study, we propose a novel configuration based on a dual-cavity fiber Fabry–Pérot interferometer (DCFFPI) that can simultaneously measure the thermo-optic coefficient (TOC) and thermal expansion coefficient (TEC) of a polymer. The polymer is particularly sensitive to T, not only regarding its size (length, L), but also its refractive index. The variation in the size (L) and refractive index (RI) of polymers with the change in T denotes the parameters of TEC and TOC. As mentioned previously, there is a cross-sensitivity between TEC and TOC in polymer interferometers because the interference wavelength shifts as n or L changes with T. The DCFFPI that cascaded a polymer and an air cavity could obtain a superimposed signal of the multiple interferences. With the proposed sensor structure, the polymer expands and compresses the air cavity when T increases. By analyzing the individual optical spectra of the polymer and the air cavity, the parameters of TEC and TOC could be determined at the same time. Experimental results indicating the accuracy in the simultaneous measurement of the TEC and TOC with small measured errors are shown to verify the feasibility and the usefulness of the proposed DCFFPI.

## 2. Sensor Fabrication and Principle

The configuration of the dual-cavity fiber Fabry–Pérot interferometer (DCFFPI) is based on a polymer and an air cavity formed as a dual cavity, as presented in Figure 1. The fabrication process is described as follows. A hollow-core fiber (HCF_1_) with an air-core diameter (D) of 50 μm was fusion-spliced to another HCF_2_ with D = 10 μm. The endface of HCF_1_ was cleaved for a certain length (preferably <150 μm) with a slight slant and followed by being spliced with a single-mode fiber (SMF). The fusion splicing could create a miniature tilted gap (2–3 μm) for the effective access of different monomers (polymers). A capillary action was used to fill the polymer into the core of HCF_1_ without fully filling the core. Filling through capillary action only took seconds to accomplish the desired structure, as there was an open cavity at the HCF_2_ endface. This provided a high fabrication efficiency. Here, R_1_, R_2_, and R_3_ denote the Fresnel reflections from the interfaces of the structure. Light that propagates into the long HCF_2_ section with a hollow core of 10 μm could not cause Fabry–Pérot interference, but would only excite some leaky modes, increasing the loss associated with the reflection of R_3_ and reducing the visibility of the interference from the air cavity [4].

Figure 2 shows the micrographs of the proposed DCFFPIs with various lengths of the dual cavity. Figure 2a,b shows the HCF_1_ of ~86 μm filled with the polymer NOA65 and HCF_1_ of ~110 μm filled with the polymer NOA61, respectively. As mentioned previously, the monomer (polymer) was added to the HCF_1_ and formed a controlled microcavity; after the filling step, the HCF_1_ was exposed to UV light. In the UV-curing process, the monomer gradually transformed into a robust polymer cavity. The remaining unfilled section was the air cavity. It is worth noting the tiny air core of the HCF_2_ as an opening for airflow.

It can improve the polymer filling and not inhibit the thermal expansion of the polymer, but it may slightly deform during the arc-discharging of the fusion splicing of fibers. Even so, the air core deformation condition did not affect the measurement results as long as HCF_2_ was not closed. The experimental setup for the simultaneous measurement of TOC and TEC of a polymer is displayed in Figure 3. The proposed DCPFFI was positioned on a TE cooler for varying T with a fixed relative humidity in the surrounding environment. When a broadband light source (BLS, BLS-GIP Technology, New Taipei, Taiwan) with an optical circulator was propagated to the device, spectral interference reflections from the polymer and air cavities were superimposed and measured by an optical spectrum analyzer (OSA, Advantest Q8381 A, Las Vegas, NV, USA). The analysis of the optical responses of the combined interferences was accomplished using the fast Fourier transform (FFT) method, which is used to separate multiple interferences in spatial frequency into two individual spatial frequencies for a polymer and air cavities [12]. Once individual interference spectra for the corresponding cavity of the polymer and air were determined, we could study the thermal effect on the characteristics of a polymer in the DCFFPI. As can be seen in Figure 1, when the DCFFPI sensor was heated by the TE cooler, the polymer cavity expanded and generated a variation in the optical length (i.e., optical path) to shift the interference wavelengths. Meanwhile, the air cavity was compressed to reduce its optical path, with the wavelength shifting to a shorter wavelength region (blue-shifted). Here, the TEC of a silica fiber ($\alpha =+5.5\times 10^{-7}\;{{}^{\circ}C}^{-1}$) was an extremely small value that could be ignored. Therefore, the length of HCF_1_ can be regarded as almost fixed.

## 3. Experimental Results and Discussions

In the experiment, the T of the proposed DCFFPI device was controlled by a TE cooler with steps of 1 °C. Several polymers of the Norland Optical Adhesive (NOA) series by Norland Products Inc., e.g., NOA61, NOA65, and NOA146 [14] were utilized for the simultaneous measurement of their TOC and TEC. Figure 4 shows the results for the measurement of the NOA61. The original spectra obtained by the OSA under T = 20~30 °C are shown in Figure 4a. Figure 4a plots the superimposed spectra from the air and polymer cavities. The FFT method was utilized to separate multiple interferences in the spatial frequency (Figure 4b) into two individual spatial frequencies for the corresponding air and polymer cavities, respectively (as shown in Figure 4c,d). In this case, the polymer length (L = 26 μm) and air length (d = 85 μm) were measured, and Figure 5 shows the experimental results of the wavelength shifts due to the variation in T.

Figure 5a,b displays the optical spectra interferences of the proposed DCFFPI for the air and polymer cavities, respectively. The interference fringes were estimated to blue-shift, shifting to the shorter wavelength region, and red-shift, shifting to the longer wavelength region, for the air and polymer cavities, respectively, due to the increase in T. It can be seen that the polymer expanded to compress the air cavity from the heating process. Heating performances with high linear sensitivities of −0.4 nm/°C and +1.0057 nm/°C were shown for the air and NOA 61-polymer cavity, respectively. Based on the above results in Figure 5a,b, the TOC and TEC can be simultaneously determined by the description below.

When the DCFFPI was heated by the TE cooler, the wavelength shifts per °C of the air (∆λ_a_) and polymer (∆λ_p_) cavities could be respectively estimated by using the following simultaneous Equations (1) and (2).
(1)$$\frac{{\Delta \lambda }_{a}}{\lambda_{a}}=\frac{\Delta d}{d}\;\;\;\;\;\;\;\;\;\;\;\;\;\left(air\;cavity\right)$$
(2)$$\frac{{\Delta \lambda }_{p}}{\lambda_{p}}=\frac{\Delta n}{n}+\frac{\Delta L}{L}+\frac{\Delta n}{n}\cdot \frac{\Delta L}{L}=\mathrm{TOC}+\mathrm{TEC}\;\left(polymer\;cavity\right)$$
where $\lambda_{a}$ and $\lambda_{P}$ are the monitored interference wavelength dip at a specific T for the air and polymer cavities (generally set at 1550 nm), respectively. The symbols d, L and Δd, ΔL denote the original values and variations in the cavity length for the air and polymer, respectively, and n is the RI of the polymer. $\frac{\Delta n}{n}\cdot \frac{\Delta L}{L}$ is extremely small that is neglected. Here, in the proposed structure, ΔL=−Δd, and the HCF_1_ can be considered essentially fixed. Based on the experimental results, the wavelength shifts from 20 °C to 30 °C for the air (∆λ_a_) and (b) polymer (∆λ_p_) cavities are shown in Figure 5a,b, respectively. Then, the TEC and TOC of the test polymer, NOA61 could be calculated by Equations (1) and (2). The results are displayed in Figure 5c and indicate that the values of the TOC and TEC of NOA61 were −1.86 × 10^−4^ (°C^−1^) and +8.3 × 10^−4^ (°C^−1^), respectively.

Figure 6 shows the results of the NOA65 polymer measurement. The results in Figure 6a,b also show the high linear sensitivity of −0.817 nm/°C and +1.563 nm/°C for the air and NOA 65-polymer cavity with cavity lengths of 59.8 μm and 26.5 μm, respectively. The TOC and TEC of NOA 65 could be simultaneously determined to be −1.9 × 10^−4^ (°C^−1^) and +11.9 × 10^−4^ (°C^−1^), respectively. Again, based on the results shown in Figure 7, thermal sensitivities of −0.64 nm/°C and +1.1147 nm/°C for the air and NOA146H polymer with cavity lengths of 65.1 μm and 24.86 μm were obtained, respectively. The simultaneous measurement of TOC and TEC yielded −3.59 × 10^−4^ (°C^−1^) and +10.78 × 10^−4^ (°C^−1^), respectively.

The above-determined TOCs and TECs obtained by the proposed approach are listed in Table 1 and compared with those in the reference data. We can see that the results of the obtained TOCs showed consistency. However, the values of the obtained TECs in this study were several times higher than those of the reference data. It can be easily understood that, in this study, with the polymer filled inside the cylindrical hollow fiber and adhered to the inner wall of the HCF, the radial expansion was almost zero to enhance the axial expansion of the polymers. Therefore, volume expansion degenerated into linear expansion to show that the linear expansion was further exacerbated. The volume expansion coefficient (γ) was demonstrated to be almost three times the linear expansion coefficient (α), i.e., γ = 3α [20]. The deviations in the TOC and TEC measurement via the proposed method could be estimated by the following relations: TOC_d_ = |TOC_m_ − TOC_r_| and TEC_d_ = |TEC_m_ − TEC_r_|, respectively. TOC_d_, TOC_m_, and TOC_r_, and TEC_d_, TEC_m_, and TEC_r_ are, respectively, the deviated, measured, and referred to values for the TOC and TEC. Here, we estimated that the TOC_d_ and TEC_d_ (linear) were 6 × 10^−6^ (°C^−1^) and 3.67 × 10^−5^ (°C^−1^) for the polymer NOA61, and 7 × 10^−6^ (°C^−1^) and 1.46 × 10^−4^ (°C^−1^) for NOA65, respectively. Consistent results show that, by using the proposed sensing scheme with a simple mathematical method, the proposed DCFFI could simultaneously and effectively sense the two parameters of TOC and TEC. We believe that the small deviations could be attributed to the measured error of the polymer/air cavity limited by the resolution of the microscope. It should be noted that, at the time of the writing of this paper, the NOA146H is a newly developed polymer; thus, there is no TOC and TEC-related information on the official website or other published references.

## 4. Conclusions

This study demonstrated a new sensing configuration based on a dual-cavity fiber Fabry–Pérot interferometer (DCFFPI) for simultaneously measuring the thermo-optic coefficient (TOC) and thermal expansion coefficient (TEC) of a polymer. The polymer is naturally highly responsive to temperature (T); thus, the variation in the length (L) or refractive index (n), or both, can generate an optical path difference to shift the interference spectra. The proposed DCFFPI with a cascaded polymer and an air cavity can resolve the cross-sensitivity between the TOC and TEC. By analyzing the individual optical spectra of the polymer and air cavities, the parameters of TOC and TEC could be determined simultaneously. A set of experimental results showed the simultaneous measurement of the TOC and TEC with small measured deviations of 6 × 10^−6^ (°C^−1^) and 3.67 × 10^−5^ (°C^−1^) for the polymer NOA61 and 7 × 10^−6^ (°C^−1^) and 1.46 × 10^−4^ (°C^−1^) for NOA65. Both the feasibility and effectiveness of the proposed DCFFPI sensor have been fully demonstrated for the class of adhesive-based polymers. We believe that the dual-cavity fiber Fabry–Pérot interferometer presented in this study can be further applied in the simultaneous measurement of the thermal optical and thermal–mechanical parameters of various materials with liquid, adhesive, or gel types. It is particularly worth mentioning that, with the proposed DCFFPI sensing mechanism, only a picoliter volume of the materials to be measured is required. This is a great merit in the investigation of precious and rare materials.

## Figures and Tables

**Figure 1 polymers-14-04966-f001:**
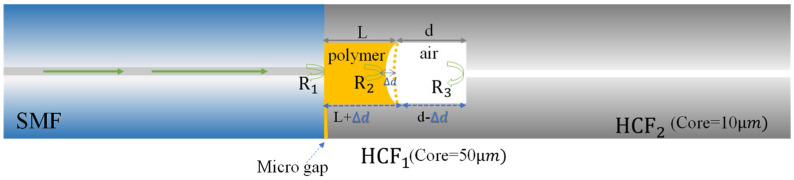
Configuration of the proposed DCFFPI with a dual cavity (polymer and air). Here, R_1_, R_2_, and R_3_ denote the Fresnel reflections from the interfaces of the structure.

**Figure 2 polymers-14-04966-f002:**
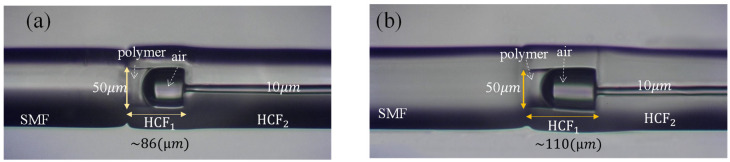
Micrographs of the proposed DCFFPIs with various structures with lengths of (**a**) HCF_1_ = 86 μm and (**b**) HCF_1_ = 110 μm. Here, HCF_1_: (D = 50 μm) and HCF_2_: (D = 10 μm).

**Figure 3 polymers-14-04966-f003:**
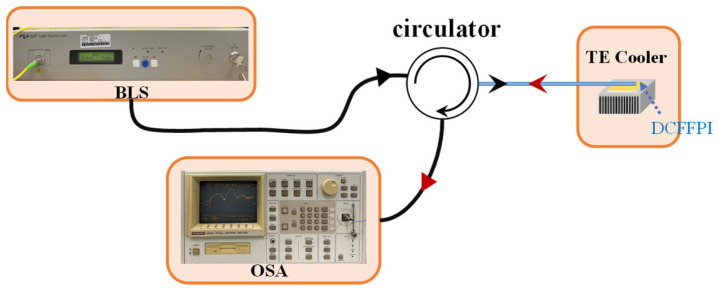
Experimental setup for simultaneously measuring the TEC and TOC of a polymer.

**Figure 4 polymers-14-04966-f004:**
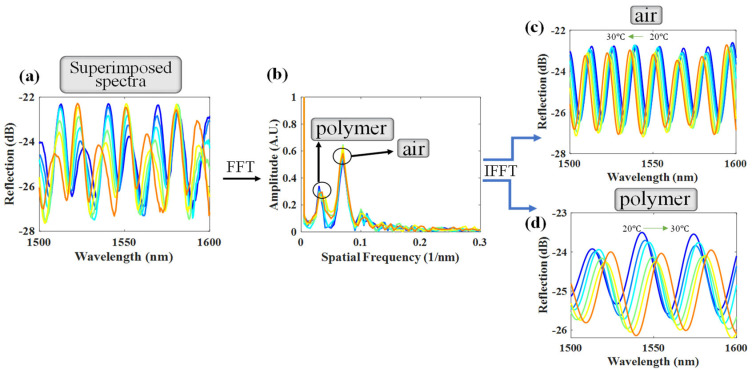
(**a**) Optical response of the superimposed interferences with variation in T; (**b**) superimposed spectra processed by FFT; separated reflection spectra of the (**c**) air cavity and (**d**) polymer cavity.

**Figure 5 polymers-14-04966-f005:**
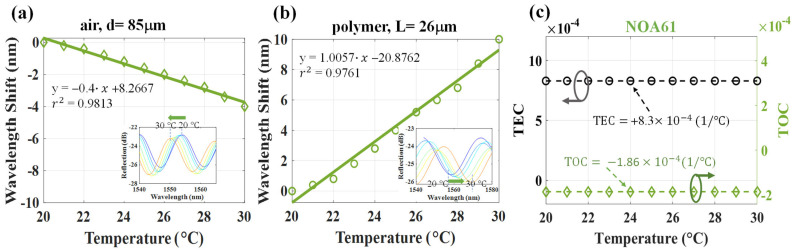
Wavelength shifts from 20 °C to 30 °C for the cavities of (**a**) air, ∆λ_a_, and (**b**) NOA61-polymer, ∆λ_p_; (**c**) measured TEC and TOC from the results of (**a**,**b**).

**Figure 6 polymers-14-04966-f006:**
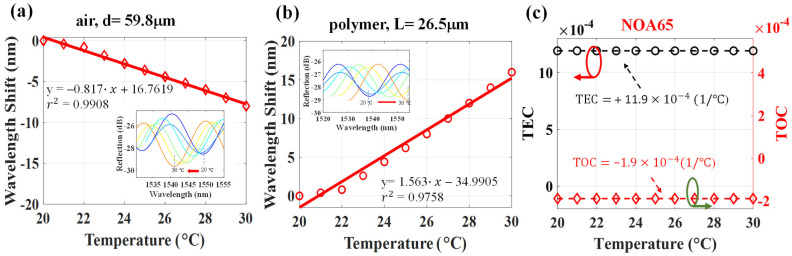
Wavelength shifts from 20 °C to 30 °C for the cavities of (**a**) air, ∆λ_a_, and (**b**) NOA65-polymer, ∆λ_p_; (**c**) measured TEC and TOC from the results of (**a**,**b**).

**Figure 7 polymers-14-04966-f007:**
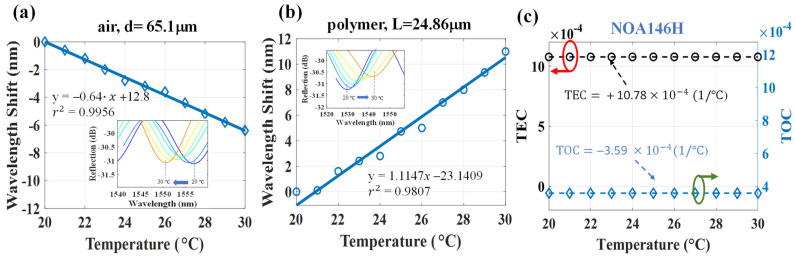
Wavelength shifts from 20 °C to 30 °C for the cavities of (**a**) air, ∆λ_a_, and (**b**) NOA146H-polymer, ∆λ_p_; (**c**) the measured TEC and TOC from the results of (**a**,**b**).

**Table 1 polymers-14-04966-t001:** Comparisons of the simultaneously measured TEC and TOC for three types of polymers between the experimental results and the reference data.

Polymers	NOA 61 (At 1550 nm)	NOA65 (At 1550 nm)	NOA146H (At 1550 nm)
TOC: (°C^−1^)	−1.86 × 10^−4^	−1.9 × 10^−4^	−3.59 × 10^−4^
TEC: (°C^−1^)	+8.3 × 10^−4^ (volume)	+11.9 × 10^−4^ (volume)	+10.78 × 10^−4^ (volume)
(measured in the study)			
TOC: (°C^−1^)	−1.8 × 10^−4^	−1.83 × 10^−4^	
TEC: (°C^−1^)	+2.4 × 10^−4^ (linear)	+2.5 × 10^−4^ (linear)	
(reference data)	[21]	[19]	
TOC_d_: (°C^−1^)	6 × 10^−6^	7 × 10^−6^	
TEC_d, linear_: (°C^−1^)	3.67 × 10^−5^	1.46 × 10^−4^	

## Data Availability

The data presented in this study are available on request from the corresponding author.

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
