# Peer review of "A Dual-Cavity Fiber Fabry–Pérot Interferometer for Simultaneous Measurement of Thermo-Optic and Thermal Expansion Coefficients of a Polymer"

_polymers, 2022, doi:10.3390/polym14224966_

Round 1

Reviewer 1 Report

The article "Two-cavity fiber Fabry-Perot interferometer for simultaneous measurement of thermo-optical and thermal expansion coefficients of a polymer" presents an interesting method for the simultaneous determination of the polymer optical and mechanical properties changes from the temperature. But there are some questions, listed below:

1. The authors argue that there are few works on the study and simultaneous measurement of the optical or mechanical parameters of polymers (lines 31-33). However, in the databases one can find quite a lot of references to such works. For example:

- Shimamura, A., Priimagi, A., Mamiya, J. I., Ikeda, T., Yu, Y., Barrett, C. J., & Shishido, A. (2011). Simultaneous analysis of optical and mechanical properties of cross-linked azobenzene-containing liquid-crystalline polymer films. ACS Applied Materials & Interfaces, 3(11), 4190-4196.

- Van Aken, J. A., & Janeschitz-Kriegl, H. (1981). Simultaneous measurement of transient stress and flow birefringence in one-sided compression (biaxial extension) of a polymer melt. Rheologica Acta, 20(5), 419-432.

- Hossain, M. F., Chan, H. P., & Uddin, M. A. (2010). Simultaneous measurement of thermo-optic and stress-optic coefficients of polymer thin films using prism coupler technique. Applied optics, 49(3), 403-408.

So, I recommend authors to make a brief review of similar papers.

 2. The method that authors proposed can be used for a limited class of polymers - polymer adhesives. This should be reflected in the abstract and conclusion of the article.

 3.  The linear thermal expansion coefficient of the polymer fiber sheath is nonzero. Please indicate how this will affect to the measurement results.

Author Response

The authors thank the reviewers for the useful suggestions and comments. Several revisions have been made in the revised version of the manuscript according to the reviewers’ comments. The changes as well as the answers to the comments can be found in the following “Reply to reviewer’s comments” (colored in blue) and in the revised manuscript (colored in red).

Reviewer 2 Report

The method proposed in the study is about the construction of an optical sensor that can measure temperature.

The expansion of the polymers at temperature resulted in a shift in wavelengths in practice.

The method applied in the study was actually combining three types of different tubes and measuring the behavior of the transmitted waves on a coupler.

The work is written fluently, practically and qualified in terms of flow and source order. I congratulate the authors in this respect.

I believe that making the corrections listed below will make the publication stand out more and attract more attention of the reader.

1-The literature of the study is somewhat weak. It is necessary to strengthen it and to emphasize the prominent parts of similar studies.

In the introduction, the contribution of the study to the literature should be emphasized about the items.

2-Paper does not have a discussion section. In the Discussion section, it would be good to present a discussion by comparing current studies that can be considered as similar or competing, by putting a table in terms of error rate, cost, etc. In this regard, I recommend examining the following studies.

https://doi.org/10.1016/j.snb.2021.130154

https://doi.org/10.1016/j.yofte.2022.102990

3- In the conclusion part, it would be good to give information about the possible usage areas of the presented method and future studies.

Author Response

(The authors gave the same response as above.)
